# Experimental Investigation of Ultrasonic Vibration-Assisted Grinding of HVOF-Sprayed WC-10Co-4Cr Coating

Ning Ji [1,2], Junhong Zhang [1,2,*], Minjie Liu [2], Huwei Dai [1,*], Kunying Ding [3], Jun Yu [4] and Xueling Zhang [2]

1 State Key Laboratory of Engines, Tianjin University, Tianjin 300072, China; jining@tju.edu.cn
2 School of Mechanical Engineering, Tianjin Renai College, Tianjin 300636, China; lijie9142@sina.com (M.L.); zhangxueling0@163.com (X.Z.)
3 School of Aeronautical Engineering, Civil Aviation University of China, Tianjin 300300, China; dingkunying@126.com
4 Aircraft Maintenance & Engineering Corporation Co., Ltd., Beijing 100621, China; yujun@ameco.com.cn
* Correspondence: zhangjh@tju.edu.cn (J.Z.); dhwmail@tju.edu.cn (H.D.)

**Abstract:** WC-10Co-4Cr coating is highly valued for its corrosion resistance and wear resistance when applied using the high-velocity oxy-fuel (HVOF) spraying method. However, conventional grinding (CG) of this coating presents challenges, including substantial grinding forces and elevated surface temperatures. To address these concerns, our study proposed the utilization of ultrasonic vibration-assisted grinding (UVAG) as a means to enhance the machining properties of HVOF-sprayed WC-10Co-4Cr coatings. Comparative experiments were conducted to analyze the impacts of various factors on the grinding forces and surface roughness in UVAG and CG processes. Additionally, the topography of the ground surfaces was examined to gain insights into the material removal mechanism in UVAG. The experimental outcomes reveal significant reductions in tangential and normal grinding forces, amounting to 15.47% and 22.23%, respectively, in UVAG when compared with CG. Furthermore, UVAG led to a roughly 29.14% decrease in ground surface roughness compared with CG. Microscopic analysis of the ground surfaces using scanning electron microscopy (SEM) indicated that ductile removal was the predominant material removal mode in UVAG. Overall, UVAG was found to be effective in diminishing grinding forces, improving ground surface roughness, and enhancing surface integrity when contrasted with CG. These findings introduce a novel approach for processing WC-10Co-4Cr coatings.

**Keywords:** ultrasonic vibration-assisted grinding; WC-10Co-4Cr coating; grinding force; surface roughness; material removal mode; ground surface topography

## 1. Introduction

  Studies show that WC-10Co-4Cr coating exhibits outstanding mechanical characteristics and can be used as a highly durable coating layer [1,2]. Accordingly, this composition has widespread applications across various industries, encompassing aerospace, automotive, and oil exploration industries. Typically, the high-velocity oxy-fuel (HVOF) method is extensively employed for the fabrication of WC-10Co-4Cr coatings [3–5]. Nonetheless, the HVOF spraying process encounters some limitations, leading to the surface roughness $R_a$ typically falling within the range of 3 to 6 μm. To meet the stringent demands for component usage, precision machining is a feasible technique used to reach the desired dimensional accuracy and surface quality [6]. Currently, the conventional grinding process faces numerous challenges when applied to this hard-to-machine material. These challenges encompass substantial grinding forces, high grinding temperatures, accelerated wear of grinding wheels, and diminished processing efficiency [7]. Consequently, it is crucial to develop innovative machining techniques to effectively reduce the grinding forces, minimize the thermal effects, and improve the surface finish. This includes methods such as ultrasound-assisted grinding, which integrates ultrasound vibrations into either the

workpiece or grinding wheel [8]; high-shear and low-pressure grinding, which employs a novel CBN liquid-body-armor-like grinding wheel with particle clustering characteristics [9]; and laser-induced ablation-assisted grinding, wherein lasers are employed to ablate the workpiece before grinding [10].

The ultrasonic vibration-assisted grinding (UVAG) requires the introduction of ultrasonic vibrations to either the workpiece or the grinding wheel while conducting the CG process. Based on experiments, UVAG affects the interplay between the abrasive particles and the workpiece. This interference in the trajectory of these particles amplifies the density of operational abrasive particles, hence diminishing the surface roughness and enhancing the surface quality [11]. Recently, UVAG has attracted numerous researchers in diverse engineering fields worldwide.

The grinding force is an influential factor in assessing grinding performance that directly influences various aspects of the grinding process, such as the temperature and efficiency of the process, and the quality of the resulting ground surface. Researchers have conducted both experimental and theoretical investigations, revealing the significant impact of ultrasonic vibration in reducing grinding forces. For instance, Yang et al. [12] conducted a study on $ZrO_2$ ceramic materials utilizing the UVAG process, revealing that $F_n$ and $F_t$ decreased by approximately 34.32% and 37.64%, respectively, in comparison with CG. They attributed this reduction to intermittent machining induced by ultrasonic vibration.

Similarly, Huang et al. [13] conducted comparative studies employing UVAG and CG processes on hardened steel. They observed reductions of approximately 16.44% in normal grinding forces and 17.44% in tangential grinding forces in UVAG. This reduction was attributed to the dynamic contact and separation of abrasive particles during one vibration cycle in UVAG, resulting in a stable cutting–separating state. Moreover, Zhang et al. [14] reported that in the CG process, the trajectory of an abrasive particle is comparatively shorter than that observed in UVAG. Additionally, the average chip thickness of an abrasive particle in UVAG is smaller than in CG, leading to lower grinding forces. They also noted that $F_n$, $F_t$, and $R_a$ in UVAG were reduced by 20%, 18%, and 9%, respectively. Furthermore, Wang et al. [15] found that elliptical vibrations in elliptical ultrasonic vibration-assisted grinding (EUAG) of monocrystal sapphire reduced $R_a$ by up to 25% compared with CG. This reduction was attributed to the continuous interaction of abrasive particles during EUAG, where they consistently cut the surface, thereby eliminating "uncut materials" between adjacent abrasive particles due to the elliptical motion. The aforementioned studies were primarily focused on factors that contribute to a decrease in grinding force by examining the contact state between abrasive particles and workpieces.

Furthermore, it should be indicated that the frequency and amplitude of ultrasonic vibration constitute two critical parameters with a substantial impact on the grinding force. By adjusting these parameters, the grinding process can be fine-tuned, thereby decreasing the grinding forces and enhancing the quality and efficiency of processing. For instance, Dai et al. [16] investigated the ultrasonic face grinding of SiC ceramic, where they employed minor vibration amplitudes. Their findings revealed that using a combination of a high-speed wheel and minor vibration amplitudes, approaching 0.2 μm, resulted in a ground surface roughness under 0.025 μm. Similarly, Yang et al. [17] conducted comparative experiments on $ZrO_2$ ceramic materials, noting that the amplitude increases as the grinding force reduces. Notably, $F_t$ and $F_n$ in the UVAG process exhibited significant reductions when compared with the CG process. Moreover, Juri et al. [18] explored ultrasonic machining in the context of diamond machining of lithium silicate ($Li_2SiO_3$ (LS)) glass ceramics. They observed that employing ultrasonic vibrations at amplitudes of 3 μm and 6 μm produces the least depth of edge chipping damage in both pre-crystallized and crystallized LS materials. This approach resulted in substantially less edge-chipping damage compared with conventional machining techniques.

Extensive research has documented wide applications of UVAG technology in the machining of hard-to-cut materials. Unfortunately, to the best of our knowledge, there have been no studies that concentrated on the use of UVAG in WC-10Co-4Cr coating,

while this process concerning WC-10Co-4Cr coating remains unexplored. The current experimental investigation intended to analyze the influence of ultrasonic vibration and grinding parameters on the grinding ability of a WC-10Co-4Cr coating when subjected to UVAG. The present paper is structured as follows: Section 2 concentrates on the analysis of the trajectory of individual abrasive particles during UVAG process. Subsequently, Section 3 describes the experimental setup and introduces the case study. Section 4 focuses on the result analysis. The primary findings and conclusions are summarized in Section 5.

## 2. Analysis of the Trajectory of a Single Abrasive Particle in UVAG

### 2.1. Trajectory of a Single Abrasive Particle

Figure 1 illustrates the schematic motion characteristics of an abrasive particle in both CG and UVAG. As can be seen in Figure 1, ultrasonic vibration is applied to the grinding wheel along the wheel axis. Additionally, Figure 1 illustrates that the trajectories of individual particles in CG and UVAG exhibit noticeable distinctions. In CG, the motion trajectory takes an arc form, while it follows a sinusoidal curve in UVAG [19,20]. To establish a coordinate system, the point at which the abrasive particle commences its penetration into the workpiece operates as the center of the coordinates. In UVAG, a single abrasive particle exhibits three motion modes, namely, high-frequency vibration, rotational movement along the wheel axis, and feed movement along the workpiece.

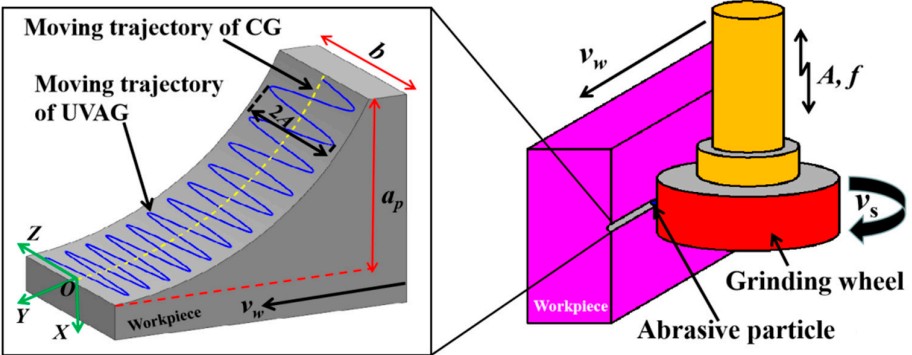

**Figure 1.** Moving trajectory of a single abrasive particle under UVAG and CG.

Under the assumption that the abrasive particles are polyhedrons of uniform size and are evenly distributed on the wheel surface at the same grinding depth, Figure 2 shows that the abrasive particle initiates cutting into the workpiece at point $A$ and exits at point $F$. This cutting action leads to the removal of material within the red-shaded area [21]. To describe the particle motion, the trajectory with respect to the motion time $t$ of a particle under UVAG is formulated as follows:

$$\begin{cases} x(t) = d_s(\cos\theta - 1)/2 \\ y(t) = -(v_w t + d_s \sin\theta/2) \\ z(t) = A\sin(2\pi f t + \phi) \end{cases} \tag{1}$$

The rotation angle $\theta$ can be obtained using the following expression:

$$\theta = \frac{2v_s t}{d_s} \tag{2}$$

The particle speed components during UVAG can be obtained as the derivative of Equation (1) with respect to time $t$:

$$\begin{cases} v_x(t) = -v_s \sin(2v_s t/d_s) \\ v_y(t) = -[v_w + v_s \cos(2v_s t/d_s)] \\ v_z(t) = 2A\pi f \cos(2\pi f t + \phi) \end{cases} \tag{3}$$

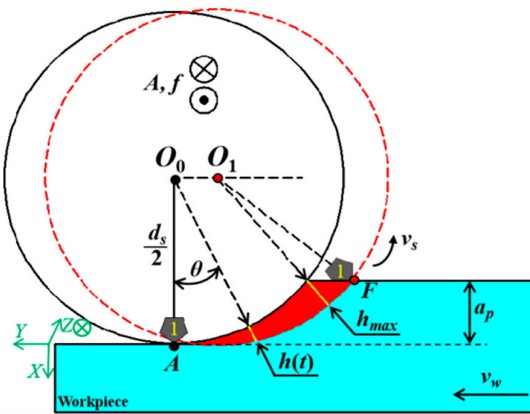

**Figure 2.** Moving schematic of a single abrasive particle under UVAG.

Similarly, the components of acceleration of the particle can be obtained from the derivative of Equation (3), as follows:

$$\begin{cases} a_x(t) = (-2v_s^2/d_s)\cos(2v_s t/d_s) \\ a_y(t) = (2v_s^2/d_s)\sin(2v_s t/d_s) \\ a_z(t) = -4A\pi^2 f^2 \sin(2\pi f t + \phi) \end{cases} \tag{4}$$

When the vibration amplitude $A$ is equal to zero ($A = 0$), Equations (1), (3), and (4) effectively represent the motion trajectory, speed, and acceleration equations of a particle in CG, respectively. As indicated by Equation (1), under the influence of ultrasonic vibration, the particle undergoes displacement along the Z-axis. This leads to a broader trajectory for an abrasive particle in UVAG, which enhances the material removal rate.

### 2.2. Interference of Abrasive Particle Trajectories in UVAG

Grinding is the cumulative result of numerous abrasive particles simultaneously cutting the workpiece materials. Because of the axial ultrasonic vibration applied, the interactions between the trajectories of abrasive particles become more intricate, with the most significant influence arising from the interference between adjacent particles. Typically, the amplitude of the ultrasonic vibration applied along the wheel axis falls within the range of 5–25 μm, which is considerably smaller than the axial spacing between axial particles on the grinding wheel [22]. As a result, the interaction between axial particles is negligible. The present study focused on the interactions between two adjacent particles situated on the circumference of the grinding wheel. Figures 3 and 4 depict the projection interference trajectories between adjacent abrasive particles on the circumference of the grinding wheel, illustrating their movement along the Z- and X-directions, respectively.

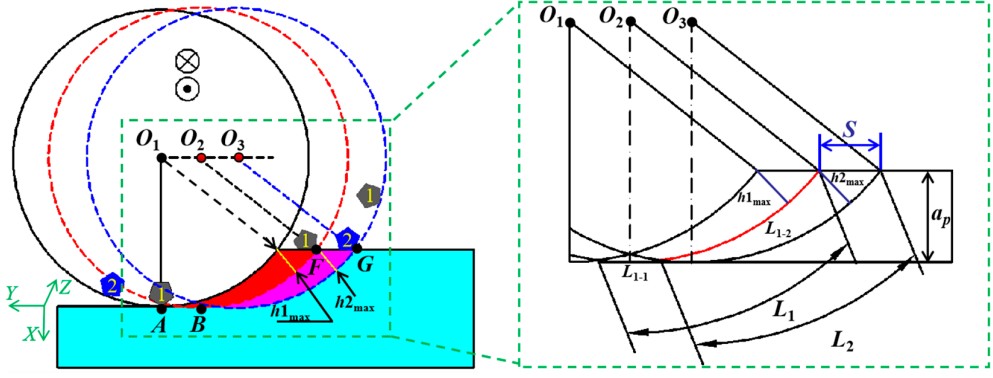

**Figure 3.** Adjacent abrasive particle projection interference trajectory along the Z-axis.

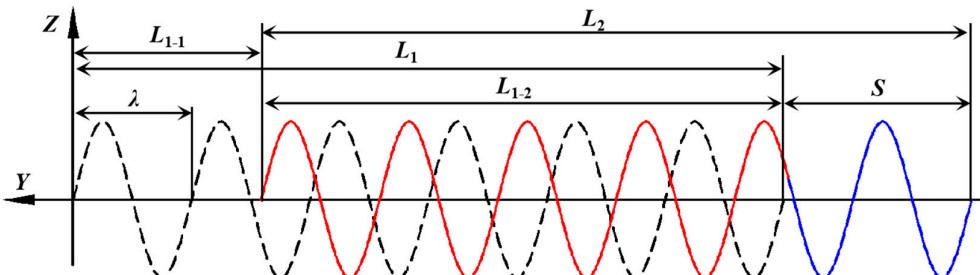

**Figure 4.** Adjacent abrasive particle projection interference trajectory along the X-axis.

The distance traveled by an abrasive particle during a complete vibration cycle, which is often referred to as one wavelength, can be expressed as

$$\lambda = \frac{v_s + v_w}{f} \tag{5}$$

The axial cutting arc length of particles can be calculated as follows:

$$L_1 = L_{1-1} + L_{1-2} \tag{6}$$

The interference length of the particle trajectory before and after it interacts with the grinding wheel can obtained from the following expression:

$$L_{1-2} = \sqrt{\left(L_2 - \sqrt{S^2 - (h_{2\max})^2}\right)^2 + (h_{2\max})^2} \tag{7}$$

The number of interference vibration cycles of two adjacent abrasive particle trajectories along the wheel axis can be calculated using the following expression:

$$n = \frac{L_{1-2}}{\lambda} \tag{8}$$

Since the trajectory of the particle interferes twice, the actual number of trajectory interferences is

$$m = 2n \tag{9}$$

The aforementioned analysis demonstrates that the number of interferences in the trajectory of particles is dependent on various factors, including wheel parameters and grinding conditions. These interferences transform the material removal process into intermittent cutting. This intermittent cutting phenomenon reduces the grinding forces, enhances the dissipation of grinding heat, and facilitates the formation of swarf. Consequently, the interference in the trajectory of abrasive particles can be enhanced to improve the overall quality of the grinding process.

## 3. Experimental Procedures

### 3.1. Specimens

In this research, 304 stainless steel specimens measuring $60 \times 40 \times 5$ mm were utilized as the substrate material. Prior to the thermal spraying process, the specimens underwent a degreasing procedure using an alkaline water-based degreaser. Additionally, the surface of the substrate was subjected to sandblasting using $Al_2O_3$ abrasive particles with an average size of 0.25 mm. This sandblasting operation was conducted under an air pressure of 0.4 MPa to enhance the adhesion between the coating layer and the surface of the workpiece. This study used commercial WC-10Co-4Cr powders (TAFA 1350 VM, Praxair Co., Ltd., Danbury, CT, USA) characterized by particle sizes falling within the range of 15–45 μm. For the spraying process, a high-pressure HP/HVOF spray gun from Praxair-TAFA was employed, which utilizes the radial powder-feeding method. This method

ensures better particle melting and an even distribution throughout the flame, resulting in improved coating coverage, efficiency, and quality. The combustion process generates particle velocities ranging from 1000 to 1185 m/s. To ensure precise control of the spraying process, the spray gun was mounted on a six-axis robot. The fuel used was aviation kerosene, oxygen served as the combustion-supporting gas, and nitrogen was employed as the powder carrier gas. The flow rate of aviation kerosene was set at 22 L/h, while the flow rate of oxygen was 50.94 m$^3$/h. The parameters for spraying included a pressure of 0.65 MPa, a horizontal moving speed of the spray gun at 400 mm/s, and a spraying step of 4 mm. The spray distance was maintained at 380 mm, and after ten rounds of the spray gun, a coating layer of 300 µm thick was obtained. For reference, a schematic depicting the HVOF process and the coating preparation equipment is presented in Figure 5.

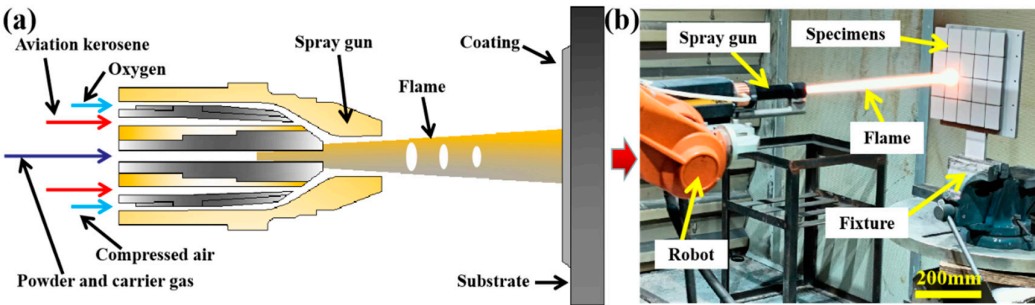

**Figure 5.** Coating preparation method: (**a**) schematic of HVOF and (**b**) coating preparation.

In the constant temperature laboratory, the mechanical properties of WC-10Co-4Cr coating were tested. The mechanical properties of the WC-10Co-4Cr coating at a constant temperature of 25 °C are detailed in Table 1.

**Table 1.** Coating characteristics at 25 °C.

| Property | Unit | Value |
|---|---|---|
| Hardness (Vickers) | $HV_{0.3}$ | 1257.1 |
| Density | kg/m$^3$ | 14,400 |
| Longitudinal Young's modulus $E_{f1}$ | GPa | 199 |
| Transverse Young's modulus $E_{f2}$ | GPa | 77 |
| Longitudinal shear modulus $G_{f12}$ | GPa | 102 |
| Transverse shear modulus $G_{f23}$ | GPa | 77 |
| Poisson's ratio $v_{12}$ | - | 0.23 |
| Poisson's ratio $v_{23}$ | - | 0.23 |
| Fracture toughness $K_{IC}$ | MPa·m$^{1/2}$ | 4.5 |

### 3.2. Grinding Equipment and Conditions

Figure 6 illustrates the experimental setup for the grinding experiments. Grinding tests were conducted using a vertical machining center (Johnford VMC-850, Taiwan, China), which boasts a highest spindle speed of 8000 rpm and a table feed speed of up to 5000 mm/min. Prior to commencing the grinding tests, the dynamic balance of the spindle was adjusted to meet the G0.4 dynamic balance standard. In these tests, an ultrasonic vibration equipment setup was employed, consisting of an ultrasonic transducer, ultrasonic generator, and ultrasonic tool holder. The ultrasonic generator converts ordinary electrical signals into high-frequency AC electrical signals, which are subsequently received by the ultrasonic transducer to generate vibration. This vibration is then transmitted to the ultrasonic tool holder, ultimately achieving tool vibration. The utilized grinding wheel had dimensions of 60 mm in diameter and 10 mm thick and featured abrasive particles with a size of 200#. The material of the abrasive was cubic boron nitride. The particle concentration in the grinding wheel was 100%, which was cooled using a commercial coolant (Csatrol Syntilo 2000, Berkshire, UK). To ensure that the test results remained unaffected by

the condition of the grinding wheel, the wheel was dressed using a trimmer before each grinding test. The trimmer was affixed to the fixture. To mitigate the influence of random factors and enhance the precision of the experimental outcomes, the process was repeated three times for each set of conditions and the average value was recorded.

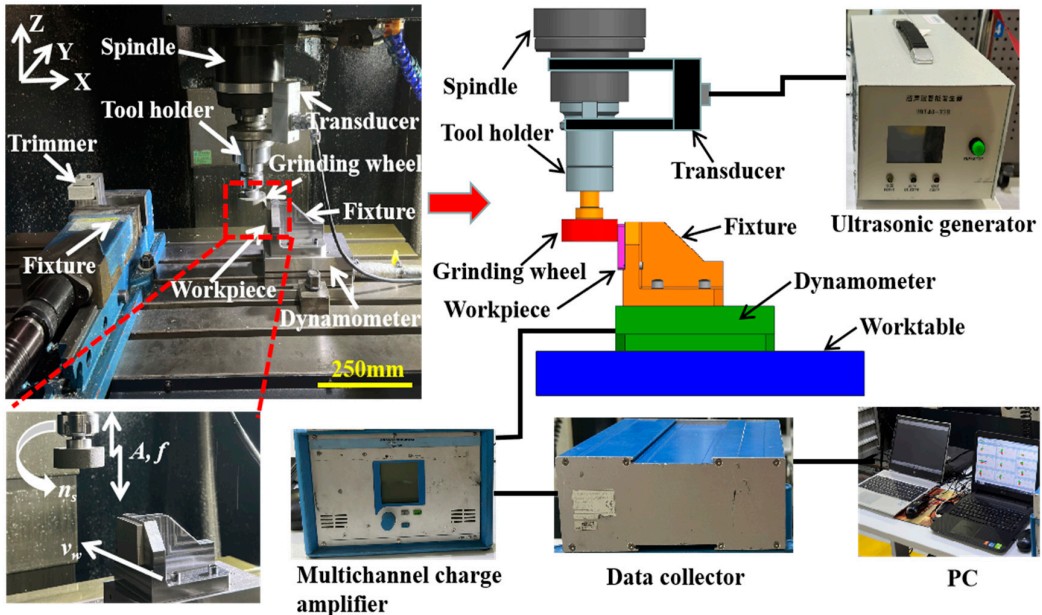

**Figure 6.** Experimental setup for the UVAG experiment.

The grinding experimental parameters are presented in Table 2.

**Table 2.** Grinding experimental parameters.

| Spindle Speed $v_s$ (m/s) | Depth of Cut $a_p$ (µm) | Feed Rate $v_w$ (mm/min) | Amplitude $A$ (µm) | Frequency $f$ (kHz) |
|---|---|---|---|---|
| 10, 14, 18, 22 | 20 | 240 | 6 | 19.8 |
| 18 | 10, 20, 30, 40 | 240 | 6 | 19.8 |
| 18 | 20 | 120, 240, 360, 480 | 6 | 19.8 |
| 18 | 20 | 240 | 6, 8, 10, 12 | 19.8 |

### 3.3. Measurement and Analysis

During the experiment, the measurement of the grinding forces was carried out using a three-phase piezoelectric dynamometer (Kistler 9257B, Winterthur, Switzerland) in conjunction with a multi-channel charge amplifier (Kistler 5070, Winterthur, Switzerland). The force measurement range for the X-, Y-, and Z-directions was configured to cover the 0~500 N range, with a sampling frequency of 10 kHz. The charge sensitivity in all three directions was set at 4.0 pC/N. Before conducting the experiments, an impedance analyzer (Sino Sonics PV520A-V, Wuxi, China) was employed to assess the reliability and stability of the ultrasonic vibration system. The test results confirmed the system's integrity, with the following parameters obtained: $f_r$ = 19.8949 kHz, $f_{ar}$ = 19.9358 kHz, $R_d$ = 14 Ω, and $C_f$ = 13 nF. These parameters aligned with those set for the experiment. The admittance chart and logarithmic impedance curve displaying these results are illustrated in Figure 7.

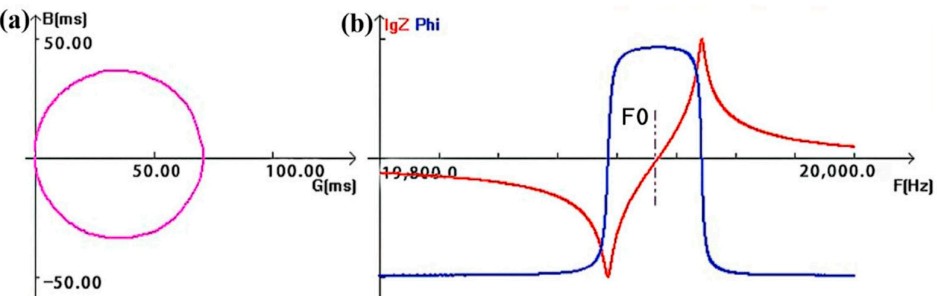

**Figure 7.** (**a**) Admittance chart and (**b**) logarithmic impedance curve.

Prior to each test, the amplitude of ultrasonic vibration was measured using a laser Doppler vibrometer. Subsequently, the power parameters of the ultrasonic generator were adjusted to ensure that the ultrasonic vibration amplitude met the specific test requirements. After completing the grinding experiments, the machined workpieces underwent cleaning with 75% alcohol using an ultrasonic cleaning process. A surface-roughness-measuring instrument (Mitutoyo SV-3100, Kangawa, Japan) was employed to measure the surface roughness of the ground samples. In order to ensure the reliability of the roughness measurement, the same surface was measured three times. The sampling length was 4 mm. The arithmetic mean of the three measurements was taken as the roughness value of the surface, and three decimal places were retained. Additionally, the grinding surface topography was examined using SEM (SNE-Alpha, Suwon, Republic of Korea).

**4. Results and Discussions**

*4.1. Grinding Force Analysis*

Grinding forces serve as a crucial indicator with significant implications for the grinding temperature, grinding wheel performance, and the quality of the ground surface [23]. In this regard, Li and Liao highlighted the utility of the $h_{\max}$ of an individual particle in determining cutting forces. This parameter can be obtained using the following expression [24]:

$$h_{\max} = \left( \frac{3 \cot \alpha}{C} \right)^{\frac{1}{2}} \left( \frac{v_w}{v_s} \right)^{\frac{1}{2}} \left( \frac{a_p}{d_s} \right)^{\frac{1}{4}} \tag{10}$$

When particles are evenly distributed, the parameter $C$ is constant, and $\alpha$ represents the half-cone angle of the abrasive particles.

The change curves depicting the variations in grinding force concerning grinding speed, feed rate, and depth of cut under the CG and UVAG processes are presented in Figure 8. It is observed that in Figure 8a, when $v_s$ rose from 10 to 22 m/s, $F_n$ and $F_t$ in CG diminished from 83.40 to 51.08 N and 37.05 to 23.56 N, respectively. There existed a similar trend in the UVAG process when compared with CG. Notably, when the grinding speed reached 18 m/s, the application of ultrasonic vibration substantially reduced the grinding force compared with CG. In this scenario, $F_n$ and $F_t$ were lower than those in the CG process by 7.33 N (12.61%) and 2.89 N (11.12%), respectively. This reduction in grinding force can be attributed to several factors: according to Equation (10), $h_{\max}$ decreases as $v_s$ increases. Consequently, the material removal volume reduces, thereby decreasing the cutting force [25]. Consequently, the final grinding force diminishes as the grinding speed rises. Under the influence of ultrasonic vibration, abrasive particles experience small displacements and accelerations on their surface. These cumulative displacements and accelerations provide additional velocity to the abrasive particles, resulting in a smaller $h_{\max}$ compared with CG. Therefore, the grinding force is reduced under UVAG in comparison with CG.

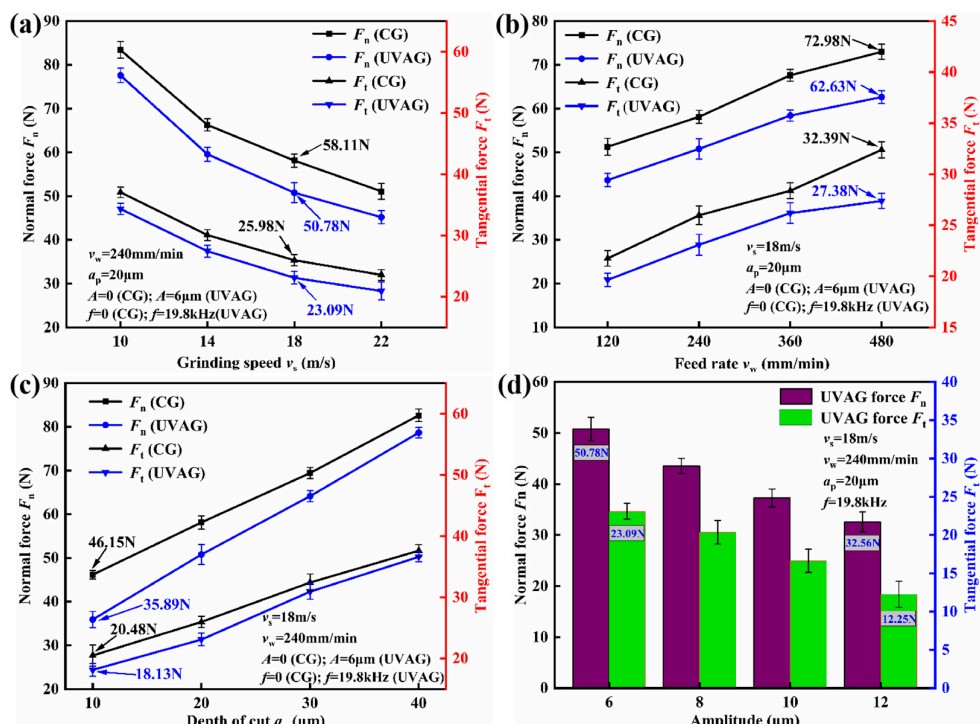

**Figure 8.** The effects of grinding and ultrasonic parameters on grinding force during UVAG and CG processes: (**a**) grinding speed, (**b**) feed rate, (**c**) depth of cut, and (**d**) amplitude of ultrasonic vibration.

Figure 8b illustrates that both $F_n$ and $F_t$ in CG and UVAG exhibited a positive correlation with $v_w$. Notably, the maximum difference between $F_n$ and $F_t$ under CG and UVAG occurred at the highest feed rate of 480 mm/min, measuring 10.35 N (14.18%) and 5.01 N (15.47%), respectively. This phenomenon was attributed to the behavior of abrasive particles. In UVAG, particles maintain contact with and separate from the workpiece within the vibration cycle, similar to a tool in a consistent cutting–separating state. Under these conditions, an increase in $v_w$ leads to intermittent grinding interactions between particles and the workpiece. This intermittent grinding effect results in a smaller grinding force when compared with the CG process.

As shown in Figure 8c, $F_n$ and $F_t$ in both CG and UVAG rose with $a_p$. Notably, the maximum difference between $F_n$ and $F_t$ under CG and UVAG occurred at an $a_p$ of 10 μm, measuring 10.26 N (22.23%) and 2.35 N (11.47%), respectively. The test outcomes demonstrate that the impact of ultrasonic vibration in reducing the grinding force weakens with increasing cutting depth. This phenomenon primarily arises because, although ultrasonic vibration enhances the self-sharpening effect of abrasive particles [26], it accelerates wheel wear when applied at a large depth of cut. This effect ultimately results in a similar level of grinding force between UVAG and CG at larger cut depths.

Figure 8d indicates that within the UVAG process, $F_n$ and $F_t$ exhibited a gradual decrease as the ultrasonic amplitude increased. For instance, when the conditions were $v_s$ = 18 m/s, $v_w$ = 240 mm/min, $a_p$ = 20 μm, and $f$ = 19.8 kHz, an increase in amplitude from 6 μm to 12 μm led to reductions of 18.22 N (35.88%) in $F_n$ and 10.84 N (46.95%) in $F_t$. This behavior primarily originates from the high-frequency vibration that abrasive particles experience due to ultrasonic vibration. This high-frequency vibration results in a sinusoidal motion track during the cutting process, as illustrated in Figure 1. The contact area between the abrasive particle and the workpiece surface increases with the increase in ultrasonic vibration amplitude, as shown in Figure 2. Consequently, the grinding force per unit area decreases, leading to a reduction in the grinding normal force $F_n$. The decrease in $F_t$ primarily originates from the anti-friction effect of ultrasonic vibration. In UVAG, abrasive particles continuously make contact with and separate from the workpiece during

the vibration cycle. As the amplitude increases, the actual contact area between particles and the workpiece diminishes, effectively reducing the friction force and diminishing the tangential force.

The performed analyses revealed that the utilization of ultrasonic vibration was an effective scheme for reducing the grinding force during the processing of the HVOF-sprayed WC-10Co-4Cr coating. The obtained results for both CG and UVAG demonstrate that both $F_n$ and $F_t$ exhibited positive a correlation with $v_w$ and $a_p$. Conversely, $F_n$ and $F_t$ exhibited a negative correlation with the $v_s$ and $A$. It is noteworthy that regardless of whether the grinding process was CG or UVAG, $F_n$ consistently surpassed $F_t$ when the same grinding parameters and vibration parameters were applied.

*4.2. Analysis of the Ground Surface Topography*

The topography of a ground surface is a crucial parameter in determining the properties of workpieces [27,28]. A well-crafted grinding surface topography can enhance the overall surface quality of the workpiece, leading to improvements in wear resistance and corrosion resistance. The topography of a ground surface is subject to the influence of various grinding processes and conditions. During the grinding process, material removal can occur through different modes, primarily involving brittle fracture and ductile removal [29,30]. These distinct material removal modes result in variations in surface topography. To gain a deeper understanding of the surface generation principles for the HVOF thermally sprayed WC-10Co-4Cr coating under UVAG, the surface topography of the coating was examined under different grinding speeds, as illustrated in Figure 9. It should be indicated that both the UVAG and CG employed identical grinding parameters, specifically $v_w$ = 240 mm/min, $a_p$ = 20 μm, $A$ = 6 μm, and $f$ = 19.8 kHz. This comparative analysis provides insights into how grinding speed impacts the surface topography for this coating.

Figure 9a,c,e,g indicate that surface defects from the CG processes were evident. These defects include pitting, scratches, fractures, smeared areas, and redeposited materials. These surface defects result from the squeezing action between the abrasive particles and the coating. The formation of long chips and large broken abrasive particles during CG leads to the creation of grinding scratches, fractures, and the redeposition of material on the surface. In contrast, Figure 9b,d,f,h indicate the surfaces after UVAG processes exhibited none of these surface defects. Moreover, there were no noticeable signs of side flow or plowing grooves on the UVAG surfaces. Several factors contribute to the improved surface quality achieved with UVAG. Ultrasonic vibration generates high-intensity mechanical vibration and induces acoustic cavitation [31,32]. The high-intensity mechanical vibration promotes the micro-crushing of abrasive particles, while the ultrasonic-induced negative pressure leads to rapid expansion and sudden bursting of micro-bubbles in the liquid through acoustic cavitation. This results in a substantial impact force, further enhancing the micro-crushing of abrasive particles. Consequently, the sharpness and dynamic stability of the abrasive particles are maintained, leading to a smoother surface finish. The application of ultrasonic vibration significantly reduces $F_t$ and $F_n$ in UVAG compared with CG. This reduction in grinding force results in decreased extrusion pressure and scraping force exerted by the particles on the coating surface. Consequently, the elastic and plastic deformation of the coating surface materials is minimized, leading to reduced overall deformation and damage on the coating surface. This ultimately contributes to the improved grinding quality in UVAG.

Figure 9a,c,e,g provide valuable insights into the effects of $v_s$ on the surface quality and material removal mode under the CG process: When the grinding speed was low ($v_s$ = 10 m/s), the coated surface exhibited more instances of pitting, fractures, scratches, and redeposited material. This suggests that brittle fracture was the primary mode of material removal under these conditions. Conversely, as $v_s$ rose to 22 m/s, the presence of pitting, fractures, and irregular area deformations on the coated surface was significantly reduced. At higher speeds, there was an increase in the plowing action of particles, indicating a

trend toward plastic deformation of the coating material. This change suggests that $h_{max}$ decreased with increasing $v_s$. Moreover, the material removal mode of the coating gradually shifted from brittle fracture to ductile removal.

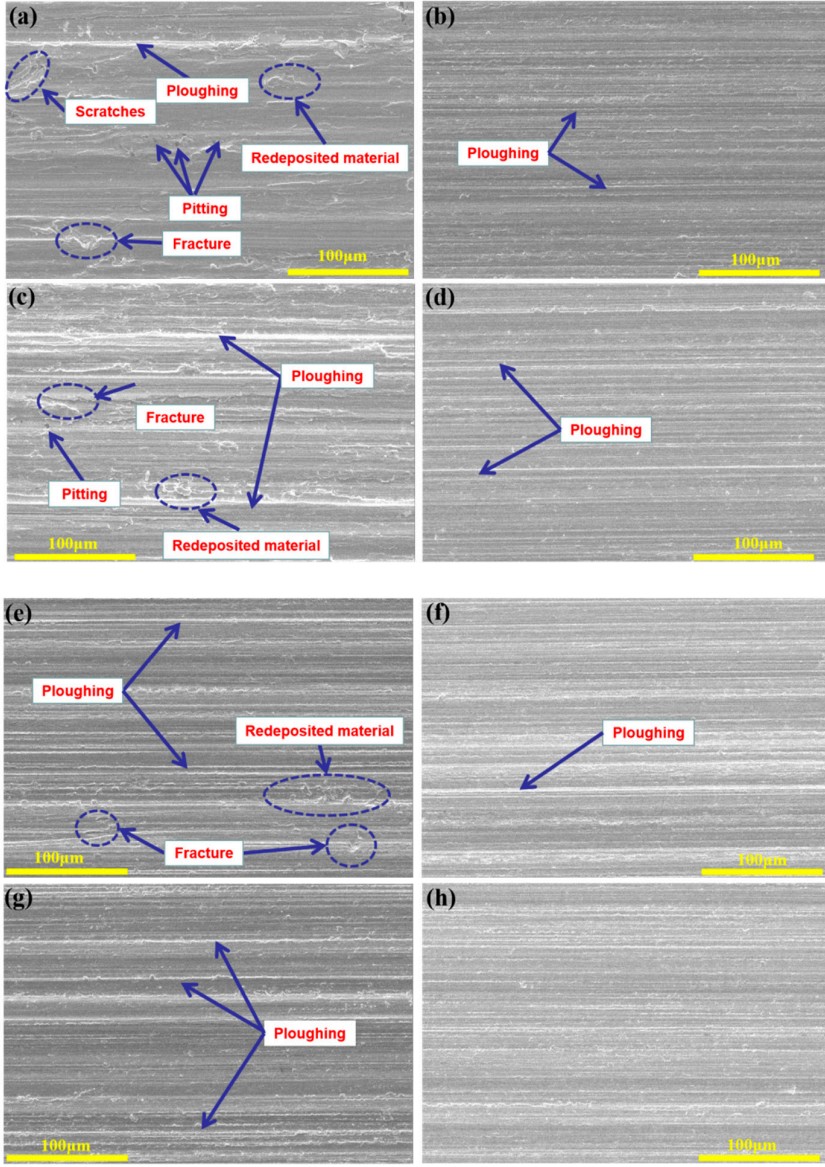

**Figure 9.** Comparison of ground surface topography between CG and UVAG at different grinding speeds: (**a**) CG $v_s$ = 10 m/s, (**b**) UVAG $v_s$ = 10 m/s, (**c**) CG $v_s$ = 14 m/s, (**d**) UVAG $v_s$ = 14 m/s, (**e**) CG $v_s$ = 18 m/s, (**f**) UVAG $v_s$ = 18 m/s, (**g**) CG $v_s$ = 22 m/s, and (**h**) UVAG $v_s$ = 22 m/s.

In contrast with the CG process, the predominant morphological feature of the grinding surface in the UVAG process was the presence of plowing grooves, as can be seen in Figure 9b,d,f,h. Importantly, this feature remained prominent even when the cutting speed was set at a relatively low value of 10 m/s. However, as $v_s$ gradually increased, significant changes occurred in the width and depth of these plowing grooves. Specifically, as $v_s$ reached 22 m/s, the grinding surface underwent a remarkable transformation. At this higher speed, the grinding surface no longer exhibited distinct plowing grooves, and the traces of these grooves became nearly imperceptible. This clear transition indicates that under UVAG, the grinding surface primarily resulted from the ductile removal mode, even when $v_s$ was set at a lower value.

### 4.3. Analysis of the Surface Roughness $R_a$

Studies show that $R_a$ is a critical factor in determining the performance of mechanical components. It substantially impacts the overall service life and reliability of mechanical products. Figure 10 presents a 2D surface roughness profile measured prior to coating grinding. The measurement results indicate that the initial average roughness ($R_a$) was 4.2 μm, and the peak-to-peak distance of the profile was 28.2 μm. These values suggest that the surface of the coating before grinding was relatively rough. Therefore, it became necessary to reduce the $R_a$ through grinding processes to enhance the surface finish. This improvement was essential to meet the requirements of components that rely on the $R_a$ of the coating.

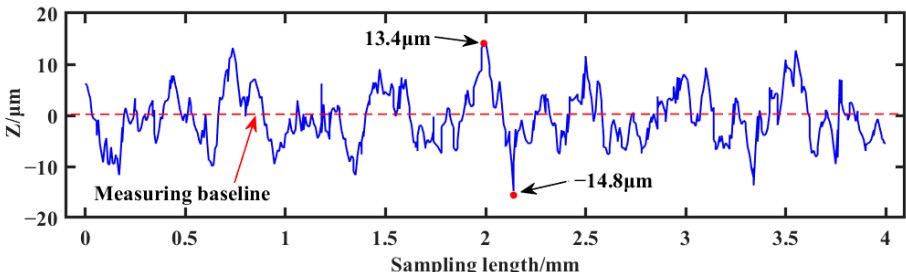

**Figure 10.** Two-dimensional surface roughness profile of coating surface prior to grinding.

Figure 11 provides an overview of the $R_a$ of the ground coating surface under various machining parameters and amplitudes for both CG and UVAG. It was observed that when $v_w$ = 240 mm/min, $a_p$ = 20 μm, $A$ = 6 μm, and $f$ = 19.8 kHz, as $v_s$ rose from 10 to 22 m/s, the surface roughness $R_a$ for both CG and UVAG decreased. Specifically, under CG, $R_a$ decreased from 0.835 μm to 0.544 μm, while under UVAG, it reduced from 0.662 μm to 0.423 μm. At $v_s$ = 22 m/s, UVAG exhibited a significantly larger reduction in surface roughness compared with CG, with a decrease of 0.121 μm, representing a 22.24% reduction. This trend aligns with the variations in grinding force and is consistent with Equation (10), where $h_{max}$ decreases as the grinding speed increases. The reduction in $h_{max}$ is crucial in decreasing the depth and width of the grinding grooves formed during the abrasive particle cutting on the coating surface. Additionally, the acceleration effect of ultrasonic vibration on abrasive particles further decreases $h_{max}$ under UVAG compared with CG, resulting in smaller grinding grooves and subsequently lower surface roughness values. This observation is consistent with electron microscope examinations.

Figure 11b reveals that when the grinding speed was set to $v_s$ = 18 m/s, and with fixed parameters such as $a_p$ = 20 μm, $A$ = 6 μm, and $f$ = 19.8 kHz, the surface roughness $R_a$ of the ground coating surface increased as $v_w$ rose from 120 to 480 mm/min. Under CG, $R_a$ rose from 0.496 μm to 1.012 μm, while under UVAG, it increased from 0.401 μm to 0.723 μm. At a feed rate of 480 mm/min, UVAG exhibited a substantial drop in surface roughness compared with CG, with a decrease of 0.289 μm, representing a 28.56% reduction. In Figure 11c, when $v_s$, $v_w$, $A$, and $f$ were held constant, the surface roughness $R_a$ was examined as $a_p$ rose from 10 to 40 μm. Under CG, $R_a$ increased from 0.477 μm to 1.004 μm, while under UVAG, it increased from 0.338 μm to 0.877 μm. At $a_p$ = 10 μm, UVAG exhibited a significant reduction in $R_a$ compared with CG, with a decrease of 0.139 μm, representing a 29.14% reduction. These trends in the surface roughness can be explained by considering $h_{max}$ as per Equation (10). An increase in $h_{max}$, which occurs with higher feed rates and deeper cuts, can lead to more of the brittle removal stage of the material, resulting in higher surface roughness. However, UVAG consistently produces smaller surface roughness values than CG under the same grinding parameters. This originates from the longer trajectory of a single abrasive particle in UVAG, as shown in Figure 1, which results in a smaller average chip thickness of the single abrasive particle in UVAG compared with CG. This leads to a smaller proportion of brittle fracture areas, ultimately reducing the surface roughness.

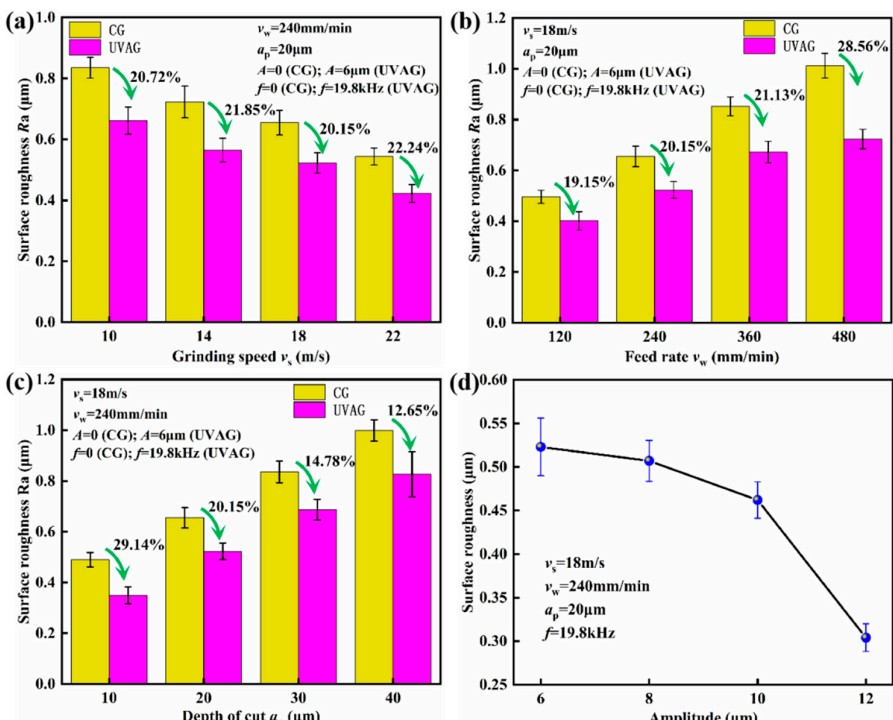

**Figure 11.** Comparison of ground surface roughness between CG and UVAG for different grinding parameters and amplitudes: (**a**) grinding speed, (**b**) feed rate, (**c**) depth of cut, and (**d**) amplitude of ultrasonic vibration.

Figure 11d shows that when $v_s$ = 18 m/s, $v_w$ = 240 mm/min, $a_p$ = 20 μm, and $f$ = 19.8 kHz, the surface roughness $R_a$ was examined as the amplitude ($A$) increased from 6 to 12 μm. Under UVAG, as the amplitude increased, $R_a$ decreased from 0.507 μm to 0.304 μm. The 2D surface roughness profiles of the grinding specimens in UVAG are shown in Figure 12, indicating that as the amplitude $A$ increased from 6 μm to 12 μm, the profile height difference decreased from 4.1 μm to 1.9 μm. This decrease in profile height difference was attributed to the "ironing" effect of abrasive particles that occurs after applying ultrasonic vibration. In UVAG, as can be seen in Figures 3 and 4, the trajectories of abrasive particles overlap and interfere with each other. Due to the uneven distribution of these particles on the grinding wheel, the same region experiences multiple rolling inter-actions with different abrasive particles. This repeated rolling creates the "ironing" effect, effectively reducing the height difference of the surface profile after applying ultrasonic vibration [33]. Furthermore, as the parameter $A$ rises, the overlap area of the abrasive particles' trajectories gradually expands, increasing the impact area of the "ironing" effect. According to Equation (4), the acceleration of abrasive particles amplifies with the increase in amplitude $A$, and thus, the particles obtain greater impact energy. This strengthens the "ironing" effect, leading to a further reduction in the surface profile height and, conse-quently, minimizing the surface roughness. This improvement in surface quality is a result of the "ironing" effect brought about by the increased amplitude in the UVAG process.

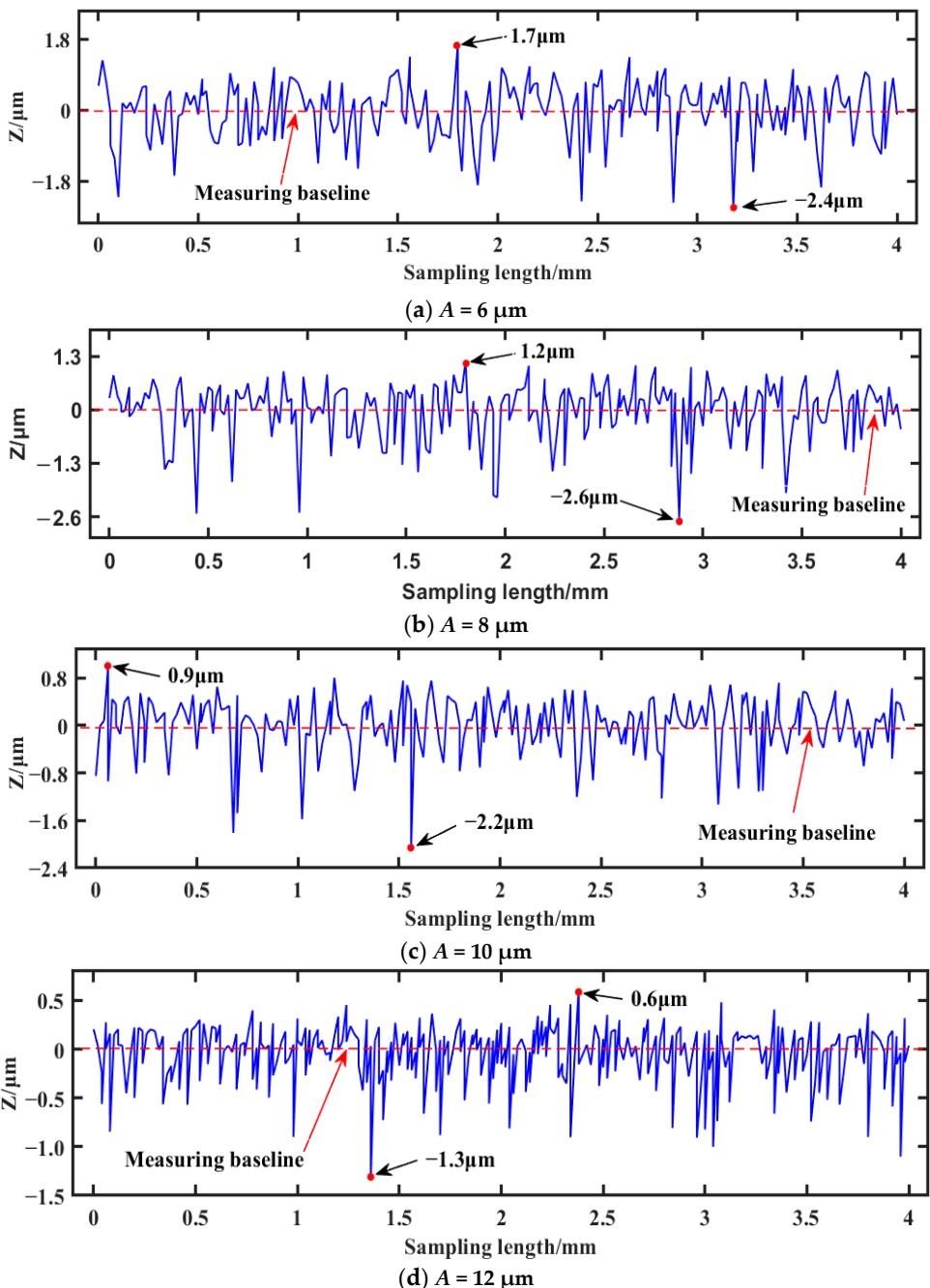

**Figure 12.** Two-dimensional surface roughness profile of coating after grinding with different ultrasonic vibration amplitude in case of $v_s$ = 18 m/s, $v_w$ = 240 mm/min, $a_p$ = 20 μm, and $f$ = 19.8 kHz.

## 5. Conclusions

This study focused on comparative experiments to investigate the grindability of an HVOF thermally sprayed WC-10Co-4Cr coating under both CG and UVAG processes. The research examined the effects of various grinding and ultrasonic vibration factors on the grinding forces, ground surface topography, and surface roughness. The main achievements of this study were as follows:

1.  UVAG resulted in lower grinding forces compared with CG, with reductions of 4.81%–22.23% for the normal force and 2.68%–15.47% for the tangential force. Additionally, the surface roughness under UVAG was found to be approximately 12.65%–29.14% lower than that under CG.

2. The grinding forces and surface roughness in the UVAG process exhibited a negative correlation with grinding speed and amplitude, while they showed a positive correlation with the worktable feed rate and depth of cut.
3. SEM observations in the UVAG process demonstrated that within the selected upper and lower limits of machining parameters studied in this research, the material removal mechanism for the HVOF thermally sprayed WC-10Co-4Cr coating exhibited a ductile domain mode.
4. The present research introduces a novel approach for processing WC-10Co-4Cr coatings.

**Author Contributions:** Conceptualization, N.J.; resources, X.Z.; methodology, K.D.; investigation, J.Z.; formal analysis, J.Z.; data curation, H.D.; writing—original draft preparation, N.J.; visualization, H.D.; writing—review and editing, M.L. and J.Y.; supervision, project administration, and funding acquisition, J.Z. All authors have read and agreed to the published version of the manuscript.

**Funding:** This project was financially supported by the Tianjin Research Innovation Project for Postgraduate Students (No. 2022KJ040).

**Institutional Review Board Statement:** Not applicable.

**Informed Consent Statement:** Not applicable.

**Data Availability Statement:** The datasets used and/or analyzed during the current study are available from the corresponding author upon reasonable request.

**Acknowledgments:** The authors would like to acknowledge the experimental assistance from Dongqing Yun at Tianjin University of Technology and Education.

**Conflicts of Interest:** The authors declare that they have no conflict of interest related to this work.

### Nomenclature

| | | |
|---|---|---|
| $A$ | Ultrasonic vibration amplitude | μm |
| $a_p$ | The cut depth | μm |
| $b$ | Grinding wheel width | mm |
| $C$ | Number of abrasive particles per unit area | particles/mm$^2$ |
| $C_f$ | Free capacitance | nF |
| $d_s$ | Grinding wheel diameter | mm |
| $F_n$ | Normal grinding force | N |
| $F_t$ | Tangential grinding force | N |
| $f$ | Ultrasonic vibration frequency | kHz |
| $f_{ar}$ | Anti-resonant frequency | kHz |
| $f_r$ | Resonant frequency | kHz |
| $h_{max}$ | Maximum undeformed cutting chip thickness | μm |
| $L_1$ | Projection length of current abrasive particles | μm |
| $L_2$ | Projection length of subsequent abrasive particles | μm |
| $R_a$ | Surface roughness | μm |
| $R_d$ | Dynamic resistance | Ω |
| $v_s$ | Grinding speed | m/s |
| $v_w$ | Feed rate of the workpiece | mm/min |
| $\theta$ | Rotation angle | deg |
| $\varphi$ | Initial phase angle of the ultrasonic vibration | deg |

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
