# Peer review of "Experimental Investigation of Ultrasonic Vibration-Assisted Grinding of HVOF-Sprayed WC-10Co-4Cr Coating"

_coatings, doi:10.3390/coatings13101788_

Round 1

Reviewer 2 Report

The authors investigated the challenges associated with conventionally grinding the WC-10Co-4Cr coating, which is known for its superior corrosion and wear resistance when applied using the HVOF spraying method. Traditional grinding of this coating leads to high grinding forces and increased surface temperatures, resulting in issues like grinding burns and inferior surface quality. The authors introduced ultrasonic vibration-assisted grinding (UVAG) as a potential solution and analysed the surface topography and roughness. The research topic presented in the manuscript is of significant interest. However, the authors must undertake a thorough revision prior to its acceptance for publication.

First, I would like to point out that the authors need to use the Journal's template. 

The introduction needs improvement. The authors need to add a few references regarding the surface topography/roughness, like the following two ( https://doi.org/10.1016/j.triboint.2022.107899 , and https://doi.org/10.1016/j.jmapro.2022.08.042 ). Additionally, the authors need to highlight the novelty of their work. 

The 2nd section should be "Materials and Methods". Basically, the authors should include additional information about the materials that they used, as well as the coating. 

The authors should enrich the "results and discussion" section by delving deeper into the analysis. It's imperative for them to juxtapose their findings with those from other relevant studies, facilitating a comprehensive understanding and contextualization of their results in the broader research.

The authors need to add more information about the instrument that they used to measure the surface roughness, how many repetitions they performed and the cutting of filters and sample length. 

Finally, the conclusions need to be improved. 

Reviewer 3 Report

It is a good manuscript. These corrections and suggestions should be made only before the acceptance.
1. The abstract should be more numerical and quantitative.
2. It is better to include equations in the text and not just in the form of equations. It means that the equation is mentioned in the text of the manuscript.
3. The number of figures is a little high. It is better to put some figures together in the text of the article.
4. It is better to use the following articles in the text of the manuscript:

* Investigation the antibacterial and photocatalytic properties of green synthesized manganese-ferrite based nanocomposite

* Hexadecyltrimethylammonium-activated and zinc oxide-coated nano-bentonite: A promising photocatalyst for tetracycline degradation

* Efficient purification of aqueous solutions contaminated with sulfadiazine by coupling electro-Fenton/ultrasound process: optimization, DFT calculation, and innovative study of …

* Heterogeneous Fenton-like Photocatalytic Process towards the Eradication of Tetracycline under UV Irradiation: Mechanism Elucidation and Environmental Risk Analysis

Reviewer 4 Report

Has good and acceptable data

Reviewer 5 Report

The manuscript focuses on the use of ultrasonic grinding for precision machining of tungsten carbide coatings. But there are a number of questions about the manuscript material:

1) In the “Introduction” section there is the phrase “only a few studies have concentrated on the use of UVAG in WC-10Co-4Cr coating while several aspects of this process concerning WC-10Co-4Cr coating remain unexplored.” What kind of research is this? What exactly has been studied previously, and what “blind spots” remain at the moment? Explain and add the necessary text.

2) Formulas (1) – (4) which model do they belong to? Where did they come from?

3) It is necessary to conduct studies of the microstructure and morphology of the resulting coating, since HVOF coatings are characterized by high porosity.

4) Does ultrasonic grinding affect the properties of the steel substrate?

5) After grinding, was the morphology of the resulting debris analyzed?

6) What abrasive material was used for grinding?

7) Fig. 12. Data is provided only for ultrasonic grinding, but is there the same profile for conventional grinding?

Round 2

Reviewer 2 Report

The Authors have improved the quality of the paper. I suggest to accept it in the present form. 

Author Response

We appreciate the valuable and constructive suggestions provided in the review reports.

Reviewer 3 Report

The manuscript is suitable for publishing.

Author Response

(The authors gave the same response as above.)

Reviewer 5 Report

It seems to me that it is necessary to carry out analyze the morphology of chips (debris) generated during the grinding process. It is especially interesting to compare chips (debris) from conventional grinding and chips (debris) from ultrasonic grinding.
